# Crowd4D: Scene-Aware Monocular 4D Crowd Reconstruction

**Hongbo Kang** [1]  **Tianyi Zhou** [1]  **Qingyang Yang** [1]  **Hongwei Wen** [1]  **Jing Huang** [1]  **Yu-Kun Lai** [2]  **Kun Li** [1]

## Abstract

Recovering scene-consistent 4D crowd motion from monocular video in large-scale scenes remains challenging due to severe depth ambiguity and complex scene geometry. Existing monocular crowd reconstruction methods typically rely on single-plane assumptions, leading to unreliable metric scale and spatial drift under complex terrain. We propose Crowd4D, the first scene-aware 4D crowd reconstruction framework that jointly optimizes the crowd and scene from a monocular RGB video in large-scale scenes. Crowd4D explicitly incorporates scene geometry and ensures consistency across image and scene spaces via a multi-stage optimization strategy. A key bottleneck of this task lies in accurate human–scene alignment, particularly in scale and position. However, human and scene reconstructions are typically decoupled. To address this, we introduce the Human–Scene Interaction Proxy (HSIP) as an intermediate representation, derived from Scene Interaction Point Clouds and a Scene Interaction Surface (SIPC&SIS), which encode explicit scene-aware geometric priors and redefine the optimization space for large-scale monocular 4D crowd reconstruction. To further improve temporal stability under occlusions, we introduce Crowd Structural Coherence Regularization (CSCR), which leverages HSIP-based spatial priors to impose soft temporal consistency on pairwise relative displacements and directions within local crowd neighborhoods. Extensive experiments demonstrate that Crowd4D consistently outperforms existing state-of-the-art methods and enables robust monocular 4D crowd reconstruction in complex, large-scale real-world scenes. Project page is available at https://cic.tju.edu.cn/faculty/likun/projects/Crowd4D.

[1]Tianjin University, Tianjin, China [2]Cardiff University, Cardiff, United Kingdom. Correspondence to: Kun Li <lik@tju.edu.cn>.

*Proceedings of the 43rd International Conference on Machine Learning*, Seoul, South Korea. PMLR 306, 2026. Copyright 2026 by the author(s).

## 1. Introduction

Monocular 4D crowd reconstruction in large scenes underpins large-scale crowd perception for urban management, public safety, intelligent transportation, and behavior understanding. In contrast to small or close-range settings, large scenes (e.g., plazas, street surveillance, drone videos) exhibit wide spatial distribution, strong scale variation, and frequent static–dynamic occlusions. With only monocular RGB input, severe depth ambiguity renders world-space localization ill-posed, often causing temporal jitter and spatial drift. Achieving spatio-temporally consistent reconstructions in both pixel and world space therefore remains the central challenge.

Existing large-scene crowd reconstruction methods (Wen et al., 2023; 2025; Huang et al., 2024) typically rely on simplified scene assumptions (e.g., single-plane terrain), which makes pixel–world consistency brittle on complex or non-planar grounds. Meanwhile, most monocular multi-person reconstruction methods are developed for small scenes: full-image approaches (Baradel* et al., 2024; Patel & Black, 2025; Wang et al., 2025b) struggle with tiny distant people, while crop-based pipelines (Goel et al., 2023; Huang et al., 2023; Wang et al., 2024) are highly sensitive to scale, where minor errors can be amplified into large world-space deviations. Depth-prior-based approaches (Liu et al., 2025; Chen et al., 2025; Allshire et al., 2025) mitigate this issue via scene reconstruction, but they mainly operate in close-range settings; for distant crowds with only a few pixels, depth ambiguity remains severe, limiting robustness in large and complex scenes.

Recent advances in scene reconstruction (Wang et al., 2025a;c) provide globally consistent geometry and accurate camera registration, offering strong geometric anchors for crowd reconstruction beyond single-plane assumptions (Wen et al., 2023; 2025; Huang et al., 2024). However, dense multi-frame point clouds are redundant and noisy, and their computational cost scales poorly with crowd size, motivating a compact and consistent interaction representation. Overall, accurate large-scale crowd reconstruction requires: (1) precise scale alignment between the crowd and the scene, (2) efficient representations that preserve multi-frame geometric consistency, and (3) robust human–scene geometric anchoring consistent with both scene geometry and image

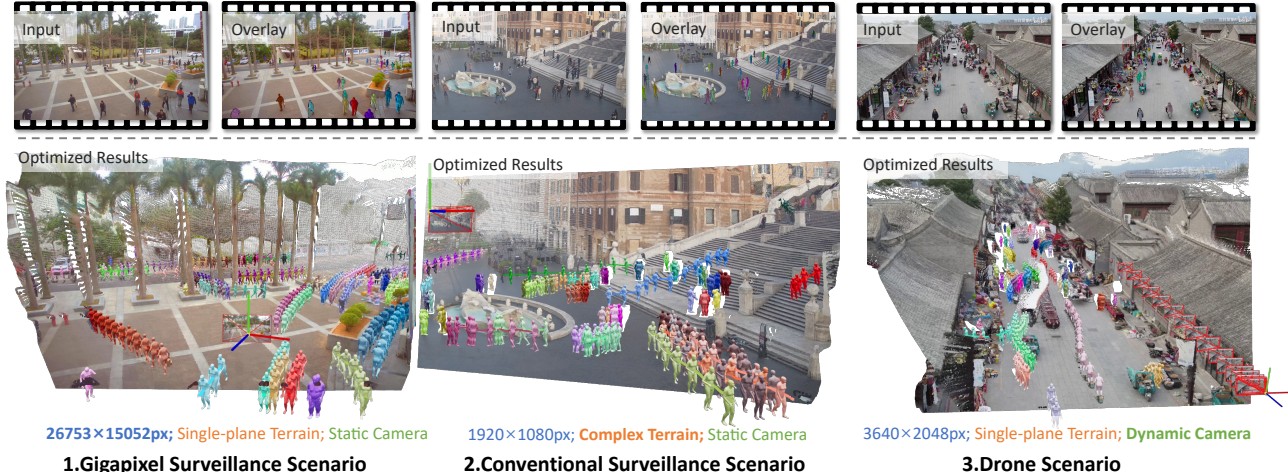

1.Gigapixel Surveillance Scenario    2.Conventional Surveillance Scenario    3.Drone Scenario

*Figure 1.* Scene-aware 4D crowd reconstruction across diverse large-scene scenarios. From left to right: a gigapixel static-camera surveillance scenario, a conventional surveillance scenario with complex terrain, and a drone-captured scenario with dynamic camera motion. Our method supports all three scenarios, whereas previous large-scene approaches are typically limited to the leftmost scenario. The top row shows the input videos and overlay results, and the bottom row presents the final optimized reconstructions. Different colors indicate different pedestrians, with per-person 3D reconstructions over time forming a scene-consistent 4D representation.

observations.

To address these challenges, we introduce the *Human–Scene Interaction Proxy* (HSIP), an intermediate optimization proxy that bridges decoupled crowd and scene reconstructions. Rather than directly constraining human poses with dense and noisy scene geometry, HSIP combines scene-level geometric priors with temporally coherent depth cues to restrict each individual to a stable and scene-consistent feasible region, mitigating the ill-posedness of monocular depth estimation. HSIP constrains each individual's world-space location while jointly optimizing the global scene scale, effectively reducing scale drift and positional instability under severe depth ambiguity. We instantiate HSIP using a compact scene representation composed of *Scene Interaction Point Clouds* and a *Scene Interaction Surface* (SIPC&SIS), which distill multi-frame reconstructions into an interaction-relevant structure. Built on SIPC&SIS, HSIP defines a feasible support region on complex terrain, enabling robust pixel-to-world anchoring beyond planar assumptions.

Moreover, per-track temporal smoothness alone is insufficient in dense crowds due to occlusions and fragmented observations. While individual motions are highly dynamic, the local structural topology of a crowd—captured by relative spatial relationships among neighbors—exhibits strong spatiotemporal coherence. Leveraging this property together with HSIP-based spatial priors, we introduce *Crowd Structural Coherence Regularization* (CSCR), a group-level constraint that enforces soft temporal consistency of pairwise relative displacements and directions, preventing implausible structural distortions under occlusion.

In summary, we propose **Crowd4D**, the first scene-aware

monocular 4D crowd reconstruction framework for large-scale complex-terrain scenes (Fig. 1). Crowd4D jointly optimizes the crowd and scene from monocular RGB video using a multi-stage strategy. Our contributions are three-fold: (1) a unified scene-aware framework for monocular 4D crowd reconstruction in large scenes; (2) HSIP, built on SIPC&SIS, which couples scene geometry and crowd distribution to stabilize metric scale and individual placement; and (3) CSCR, which exploits local crowd structure to improve temporal stability under occlusions. Extensive experiments demonstrate state-of-the-art performance on large-scale scenes.

## 2. Related Work

### 2.1. Human-Scene Reconstruction

Joint human–scene reconstruction aims to recover humans, cameras, and scene geometry in a common world frame. Sensor-assisted methods, such as CIMI4D (Yan et al., 2023) and HiSC4D (Dai et al., 2024), capture accurate human–scene interactions using wearable inertial sensors and Li-DAR, but require specialized hardware. Vision-based methods jointly reason about humans and scenes from visual observations: SynCHMR (Zhao et al., 2024) reconstructs metric-scale cameras, humans, and dense scenes from monocular videos; HSfM (Müller et al., 2024) jointly recovers humans, cameras, and scene point clouds from sparse un-calibrated multi-view images; and optimization-based methods such as JOSH (Liu et al., 2025) and VideoMimic (Allshire et al., 2025) couple human motion with reconstructed environments for scene-consistent motion recovery. However, existing vision-based methods mainly target individual humans or sparse interactions in relatively local scenes.

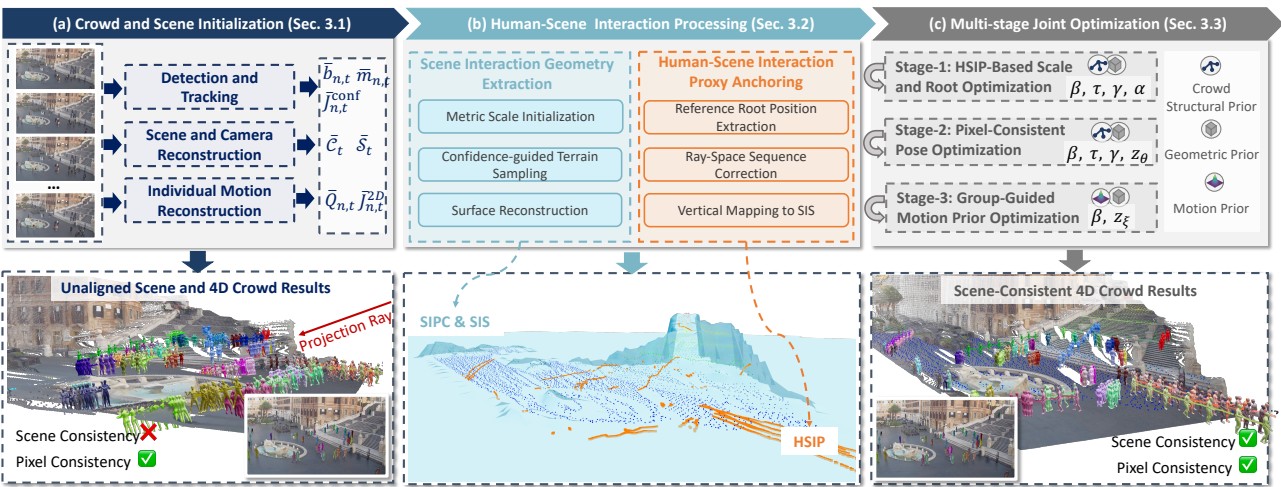

*Figure 2.* Overview of the Crowd4D framework. Our method starts from crowd and scene initialization; however, the initial results exhibit significant deviations in scene scale and individual spatial locations. To address this issue, we introduce human–scene interaction processing, where geometric constraints are imposed via Scene Interaction Point Clouds and a Scene Interaction Surface (SIPC&SIS) to construct Human–Scene Interaction Proxies (HSIP) that guide subsequent optimization. Finally, with crowd structure, geometry, and motion priors, a multi-stage joint optimization is performed to obtain scene-consistent and pixel-consistent 4D crowd reconstruction results.

In large-scale crowd videos, distant observations, severe inter-person occlusions, and complex non-planar terrain make direct human–scene alignment unreliable. In contrast, we construct compact terrain-aware interaction geometry, SIPC&SIS, and derive HSIP anchors to stabilize dense crowd placement on complex terrain.

## 2.2. Crowd Reconstruction

Multi-person methods like ROMP (Sun et al., 2021), Multi-HMR (Baradel* et al., 2024), and AiOS (Sun et al., 2024) reconstruct meshes simultaneously but focus on small scenes and cannot handle dense crowds spanning hundreds of meters. For large-scene reconstruction, Crowd3D (Wen et al., 2023) introduces HVIP (Human-scene Virtual Interaction Point) to convert 3D localization into 2D pixel estimation with pre-estimated camera and ground plane. GroupRec (Huang et al., 2023) models inter-person interactions via hypergraph reasoning. RCR (Huang et al., 2024) defines ground-aware normalization to eliminate camera intrinsic dependence. However, these single-image methods cannot infer crowd motion from video. DyCrowd (Wen et al., 2025) extends to video with group-guided optimization and asynchronous consistency loss, and it also adopts HVIP-based localization. However, HVIP-style 2D-to-3D plane anchoring is sensitive to projection errors: small 2D deviations can induce large 3D drift, especially in low-altitude, long-range scenes. In contrast, our HSIP combines scene-level geometric priors with temporally coherent depth cues to define stable feasible regions, and further stabilizes dense crowds via local structural coherence beyond per-individual temporal smoothness.

## 3. Methodology

Crowd4D aims to recover scene-consistent 4D crowd motion from a monocular RGB video by jointly estimating the global camera trajectory, static scene geometry, and the 3D positions, poses, and shapes of multiple individuals in a unified world coordinate system. To address scale inconsistency and spatial drift caused by depth ambiguity and complex terrain, we formulate crowd and scene reconstruction as a tightly coupled joint optimization problem with geometric, group-level, and motion constraints, enforcing consistency between image and scene spaces (Fig. 2).

**Human Body Representation.** We represent human geometry and motion using the SMPL model (Loper et al., 2015). At time $t$, each person $n$ is parameterized by pose $\boldsymbol{\theta}_{n,t} \in \mathbb{R}^{23 \times 3}$, shape $\boldsymbol{\beta}_{n,t} \in \mathbb{R}^{10}$, global orientation $\boldsymbol{\gamma}_{n,t} \in \mathbb{R}^3$, and global translation $\boldsymbol{\tau}_{n,t} \in \mathbb{R}^3$. We denote $\mathbf{Q}_{n,t} = (\boldsymbol{\theta}_{n,t}, \boldsymbol{\beta}_{n,t}, \boldsymbol{\gamma}_{n,t}, \boldsymbol{\tau}_{n,t})$ and the temporal sequence as $\mathbf{Q}_n = \{\mathbf{Q}_{n,t}\}_{t=1}^T$. The SMPL forward function maps these parameters to a posed mesh $\mathbf{V}_{n,t} \in \mathbb{R}^{6890 \times 3}$ and joint locations $\mathbf{J}_{n,t} \in \mathbb{R}^{24 \times 3}$ in the world coordinate system:

$$(\mathbf{V}_{n,t}, \mathbf{J}_{n,t}) = \mathcal{M}(\boldsymbol{\gamma}_{n,t}, \boldsymbol{\theta}_{n,t}, \boldsymbol{\beta}_{n,t}) + \boldsymbol{\tau}_{n,t}. \quad (1)$$

### 3.1. Crowd and Scene Initialization

We construct a globally consistent initialization to bootstrap the subsequent joint optimization, which outputs: (i) per-frame person detections with track identities, (ii) a gravity-aligned global camera trajectory and static scene point cloud, and (iii) per-person initial SMPL motion sequences in a shared world frame.

**Detection and Tracking.** Given a frame $\mathbf{I}_t \in \mathbb{R}^{H \times W \times 3}$, we apply YOLOX (Ge et al., 2021) to detect persons in the full frame. For each detected person, we obtain an instance mask using SAM 2 (Ravi et al., 2025) and per-joint observation confidence scores from DWPose (Yang et al., 2023). BoostTrack++ (Stanojević & Todorović, 2024) associates detections over time, yielding $N$ persistent identities. For each valid person–frame pair $(n, t)$, we denote the associated bounding box, instance mask, and per-joint observation confidence scores as $\bar{\mathbf{b}}_{n,t} \in \mathbb{R}^4$, $\bar{\mathbf{m}}_{n,t} \in \{0, 1\}^{H \times W}$, and $\bar{\mathbf{J}}_{n,t}^{\mathrm{conf}} \in [0, 1]^{17}$, respectively.

**Scene and Camera Reconstruction.** We uniformly sample $T_{\mathrm{key}}$ keyframes and apply $\pi^3$ (Wang et al., 2025c) to reconstruct camera parameters and scene geometry. For each keyframe $t$, we obtain camera parameters $\bar{\mathcal{C}}_t = \{\bar{\mathbf{K}}_t, \bar{\mathbf{R}}_t, \bar{\mathbf{t}}_t\}$ together with a pixel-aligned 3D point map defined in its local camera coordinate system and its corresponding confidence map, where $\bar{\mathbf{K}}_t$ is the camera intrinsic matrix, $\bar{\mathbf{R}}_t \in \mathrm{SO}(3)$ is the rotation matrix, and $\bar{\mathbf{t}}_t \in \mathbb{R}^3$ is the translation vector. We collect valid points from the predicted point map to form the per-frame scene point cloud $\bar{\mathcal{S}}_t = \{(\bar{\mathbf{s}}_j^t, \bar{c}_j^t)\}_{j=1}^{N_t^{\mathrm{scn}}}$, where $N_t^{\mathrm{scn}}$ is the number of retained points in keyframe $t$, $\bar{\mathbf{s}}_j^t \in \mathbb{R}^3$ denotes the 3D coordinate of the $j$-th point, and $\bar{c}_j^t \in [0, 1]$ is its confidence score. Finally, we estimate gravity using GeoCalib (Veicht et al., 2024) and transform all camera poses and point clouds into a first-frame-referenced, gravity-aligned world coordinate system.

**Individual Motion Reconstruction.** For each tracked identity $n$, we apply a monocular motion reconstruction method (Wang et al., 2024) to estimate temporally consistent SMPL parameters $(\bar{\boldsymbol{\theta}}_{n,t}, \bar{\boldsymbol{\beta}}_{n,t}, \bar{\boldsymbol{\gamma}}_{n,t}, \bar{\boldsymbol{\tau}}_{n,t})$. The reconstructed motion is transformed into the gravity-aligned world coordinate system to initialize $\bar{\mathbf{Q}}_{n,t}$. We further project the corresponding initialized SMPL joints, mapped to the adopted 17-keypoint layout, onto frame $t$ to obtain the 2D reference joints $\bar{\mathbf{J}}_{n,t}^{\mathrm{2D}} \in \mathbb{R}^{17 \times 2}$ for subsequent optimization.

### 3.2. Human and Scene Interaction Processing

To reduce redundancy and noise in dense scene point clouds, we extract compact terrain-aware geometry in the form of a *Scene Interaction Point Cloud* (SIPC) and a continuous *Scene Interaction Surface* (SIS), which together support the construction of the *Human–Scene Interaction Proxy* (HSIP) for coupling the scene and the crowd.

#### 3.2.1. SCENE INTERACTION GEOMETRY EXTRACTION

**Metric Scale Initialization.** Monocular reconstruction suffers from inherent scale ambiguity. We estimate a global scale factor $\alpha \in \mathbb{R}^+$ by aligning the depth of reconstructed humans with the scene geometry using the metric SMPL prior. For each valid person–frame pair $(n, t)$, we use the lowest body vertex along the gravity axis, denoted as $\bar{\mathbf{v}}_{n,t}^{\downarrow}$, as a proxy for ground contact. The initial scale $\alpha_0$ is computed as a weighted average of the ratio between human-based depth $d_{n,t}^{\mathrm{smpl}}$ and scene-based depth $d_{n,t}^{\mathrm{scn}}$:

$$\alpha_0 = \frac{1}{Z} \sum_{(n,t) \in \Omega} \omega_{n,t} \frac{d_{n,t}^{\mathrm{smpl}}}{d_{n,t}^{\mathrm{scn}}}, \qquad Z = \sum_{(n,t) \in \Omega} \omega_{n,t}, \quad (2)$$

where $\omega_{n,t}$ assigns higher weight to individuals closer to the camera. The estimated scale is applied to initialize the metric scene and camera trajectories.

**Confidence-guided Terrain Sampling.** Following metric scale initialization, we aggregate the scaled scene point clouds from all $T_{\mathrm{key}}$ keyframes into a unified gravity-aligned representation $\mathcal{S} = \{(\mathbf{s}_m, c_m)\}_{m=1}^M$. Rather than operating on the dense and noisy point cloud directly, we introduce the *Scene Interaction Point Cloud* (SIPC) as a compact terrain-aware abstraction that preserves interaction-relevant geometry while suppressing redundancy.

Specifically, we discretize the horizontal $(x, z)$ plane into a regular metric grid with cell size $\ell$. For each grid cell indexed by $\mathbf{u} = (u_x, u_z) \in \mathbb{Z}^2$, we collect scene points falling inside the cell:

$$\mathcal{G}_{\mathbf{u}} = \left\{ (\mathbf{s}_m, c_m) \in \mathcal{S} \mid \left\lfloor \frac{x_m}{\ell} \right\rfloor = u_x, \left\lfloor \frac{z_m}{\ell} \right\rfloor = u_z \right\}. \quad (3)$$

We retain only high-confidence measurements in each cell:

$$\mathcal{G}_{\mathbf{u}}^{\mathrm{conf}} = \{ (\mathbf{s}_m, c_m) \in \mathcal{G}_{\mathbf{u}} \mid c_m \geq \tau_c \}, \quad (4)$$

where $\tau_c$ denotes the confidence threshold. For each non-empty reliable cell, we select the lowest-elevation point along the gravity axis as the local terrain representative:

$$\mathbf{s}_{\mathbf{u}}^{\star} = \arg \min_{(\mathbf{s}_m, c_m) \in \mathcal{G}_{\mathbf{u}}^{\mathrm{conf}}} y_m. \quad (5)$$

The resulting set $\mathcal{S}_{\mathrm{SIPC}} = \{\mathbf{s}_{\mathbf{u}}^{\star}\}$ constitutes the Scene Interaction Point Cloud, which provides a sparse yet structurally consistent proxy of the navigable surface for subsequent human–scene interaction modeling.

**Surface Reconstruction.** To enable continuous interaction queries, we construct the *Scene Interaction Surface* (SIS) from the discrete SIPC anchors. We fit a reference support plane and interpolate the gravity-direction residuals of SIPC points over a regular horizontal grid. In insufficiently observed regions, the residual field is smoothly attenuated toward zero, allowing the reconstructed surface to fall back to the reference plane and preventing unstable extrapolation. We triangulate the resulting height field in

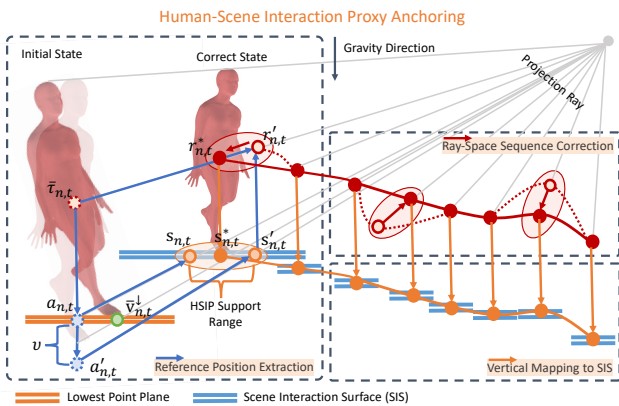

*Figure 3.* Human-Scene Interaction Proxy Anchoring.

the horizontal plane and discard triangular faces with excessively long edges to avoid unsupported connections. The resulting surface is represented as

$$\mathcal{S}_{\text{SIS}} = (\mathcal{V}_{\text{SIS}}, \mathcal{F}_{\text{SIS}}), \tag{6}$$

where $\mathcal{V}_{\text{SIS}}$ and $\mathcal{F}_{\text{SIS}}$ denote the mesh vertices and triangular faces, respectively. SIS supports gravity-direction projection and ray–surface intersection in subsequent HSIP construction.

### 3.2.2. HUMAN-SCENE INTERACTION PROXY ANCHORING

Monocular reconstruction provides reliable image-space alignment but leaves depth and support relations underconstrained. We therefore construct a scene-aware interaction proxy by progressively restricting the admissible 3D location of the human root while maintaining consistency with its original 2D projection, as shown in Fig. 3.

**Reference Position Extraction.** For each person $n$ at frame $t$, we use the lowest body vertex $\bar{\mathbf{v}}_{n,t}^{\downarrow}$ as a gravity-consistent support reference. We first define a plane anchor by vertically projecting the initialized root translation onto the horizontal plane passing through this point:

$$\mathbf{a}_{n,t} = \big(x(\bar{\boldsymbol{\tau}}_{n,t}),\, y(\bar{\mathbf{v}}_{n,t}^{\downarrow}),\, z(\bar{\boldsymbol{\tau}}_{n,t})\big). \tag{7}$$

An auxiliary probe anchor is obtained by extending it downward by a fixed offset $v$:

$$\mathbf{a}'_{n,t} = \mathbf{a}_{n,t} - (0, v, 0)^{\top}. \tag{8}$$

Given the camera extrinsics, the viewing ray corresponding to a world-space point $\mathbf{x}$ is parameterized as

$$\mathbf{r}(\mu \mid \mathbf{x}) = -\bar{\mathbf{R}}_t^{\top}\bar{\mathbf{t}}_t + \mu\big(\mathbf{x} + \bar{\mathbf{R}}_t^{\top}\bar{\mathbf{t}}_t\big). \tag{9}$$

Intersecting the rays of $\mathbf{a}_{n,t}$ and $\mathbf{a}'_{n,t}$ with the Scene Interaction Surface yields the scene correspondences

$$\mathbf{s}_{n,t} = \mathbf{r}(\mu \mid \mathbf{a}_{n,t}) \cap \mathcal{S}_{\text{SIS}}, \qquad \mathbf{s}'_{n,t} = \mathbf{r}(\mu \mid \mathbf{a}'_{n,t}) \cap \mathcal{S}_{\text{SIS}}. \tag{10}$$

Let $\mathbf{l}^{\text{vert}}(\mathbf{x})$ denote the gravity-aligned vertical line passing through point $\mathbf{x}$. We define the reference position as the intersection between the root viewing ray and the gravity-aligned vertical line passing through the probe correspondence:

$$\mathbf{r}'_{n,t} = \mathbf{r}(\mu \mid \bar{\boldsymbol{\tau}}_{n,t}) \cap \mathbf{l}^{\text{vert}}(\mathbf{s}'_{n,t}). \tag{11}$$

**Ray-Space Sequence Correction.** The reference positions $\mathbf{r}'_{n,t}$ are estimated independently per frame and may exhibit temporal jitter. We therefore parameterize them by the depth $\mu_{n,t}$ along the root viewing ray,

$$\mathbf{r}'_{n,t} = \mathbf{r}(\mu_{n,t} \mid \bar{\boldsymbol{\tau}}_{n,t}), \tag{12}$$

and apply robust temporal smoothing:

$$\widetilde{\mu}_{n,t} = \mathcal{F}\big(\{\mu_{n,\tau}\}_{\tau \in \mathcal{T}(t)}\big). \tag{13}$$

The refined reference position is obtained as

$$\mathbf{r}^*_{n,t} = \mathbf{r}(\widetilde{\mu}_{n,t} \mid \bar{\boldsymbol{\tau}}_{n,t}), \tag{14}$$

which preserves image-space alignment while yielding temporally coherent trajectories.

**Vertical Mapping to SIS.** Finally, we map the refined reference position to the scene by intersecting the gravity-aligned vertical line passing through $\mathbf{r}^*_{n,t}$ with the Scene Interaction Surface:

$$\mathbf{s}^*_{n,t} = \mathbf{l}^{\text{vert}}(\mathbf{r}^*_{n,t}) \cap \mathcal{S}_{\text{SIS}}. \tag{15}$$

We refer to $\mathbf{s}^*_{n,t}$ as the *Human–Scene Interaction Proxy (HSIP)*. To characterize its spatial extent, we define the HSIP support range as

$$r_{n,t}^{\text{HSIP}} = \max\big(\|\mathbf{s}^*_{n,t} - \mathbf{s}_{n,t}\|,\, \|\mathbf{s}^*_{n,t} - \mathbf{s}'_{n,t}\|\big). \tag{16}$$

For open- and mid-range scenes, we construct the Human–Scene Interaction Proxy (HSIP) from SIPC&SIS as described above. For close-range scenes, where foreground bodies often occlude the local support region and hinder reliable surface construction, we bypass SIPC/SIS and estimate HSIP directly in ray space. Specifically, we define an anchor ray using the image-space projection of the support-plane anchor $\mathbf{a}_{n,t}$, infer a target horizontal support location from local scene-depth observations around reliable body joints or the person mask, and determine $\mathbf{s}^*_{n,t}$ as the point on the anchor ray whose horizontal position is closest to the inferred support location. Since SIS-based correspondences are unavailable in this setting, we set the HSIP support range $r_{n,t}^{\text{HSIP}}$ to a fixed constant for close-range scenes.

### 3.3. Multi-stage Optimization Strategy

Given the reconstructed camera trajectory $\{\bar{\mathcal{C}}_t\}$, the initial human motion sequences $\{\bar{\mathbf{Q}}_{n,t}\}$, and the Scene Interaction Surface (SIS) together with the Human–Scene Interaction Proxy (HSIP), we formulate crowd reconstruction as a

| Stage | Opt. Variables | Energy Term |
|-------|----------------|-------------|
| Stage-1 | $\boldsymbol{\beta}$, $\boldsymbol{\tau}$, $\boldsymbol{\gamma}$, $\alpha$ | $\mathcal{E}_{\text{proj}} + \mathcal{E}_{\text{HSIP}}^{(proj)} + \mathcal{E}_{\text{HSIP}}^{(xz)} + \mathcal{E}_{\text{HSIP}}^{(y)}$ 
 $+ \mathcal{E}_{\boldsymbol{\beta}} + \mathcal{E}_{\boldsymbol{\tau}}^{(t)} + \mathcal{E}_{\boldsymbol{\gamma}}^{(t)} + \mathcal{E}_{\text{crowd}}^{(t)}$ |
| Stage-2 | $\boldsymbol{\beta}$, $\boldsymbol{\tau}$, $\boldsymbol{\gamma}$, $\mathbf{z}_\theta$ | $\mathcal{E}_{\text{proj}} + \mathcal{E}_{\text{HSIP}}^{(proj)} + \mathcal{E}_{\text{HSIP}}^{(y)} + \mathcal{E}_{\text{prior}}^{\theta}$ 
 $+ \mathcal{E}_{\boldsymbol{\beta}} + \mathcal{E}_{\boldsymbol{\tau}}^{(t)} + \mathcal{E}_{\boldsymbol{\gamma}}^{(t)} + \mathcal{E}_{\boldsymbol{\theta}}^{(t)} + \mathcal{E}_{\text{crowd}}^{(t)}$ |
| Stage-3 | $\boldsymbol{\beta}$, $\mathbf{z}_\xi$ | $\mathcal{E}_{\text{proj}} + \mathcal{E}_{\text{HSIP}}^{(y)} + \mathcal{E}_{\text{contact}}^{(v)} + \mathcal{E}_{\text{prior}}^{\xi}$ 
 $+ \mathcal{E}_{\boldsymbol{\beta}} + \mathcal{E}_{\boldsymbol{\tau}}^{(t)} + \mathcal{E}_{\boldsymbol{\gamma}}^{(t)} + \mathcal{E}_{\boldsymbol{\theta}}^{(t)} + \mathcal{E}_{\text{connect}} + \mathcal{E}_{\text{AMC}}$ |

*Table 1.* Overview of the multi-stage optimization strategy: variables and energy terms used in each stage.

joint optimization problem over the human motion variables $\{\mathbf{Q}_{n,t}\}$ and the residual global scene scale $\alpha$ in a unified world coordinate system, and solve it using a three-stage optimization strategy where different subsets of variables and energy terms are progressively activated to improve stability and convergence (see Table 1); the energy terms for Crowd Structural Coherence Regularization and HSIP-based Crowd–Scene Alignment are presented in Secs. 3.5 and 3.4, while the remaining terms are provided in Appendix A.

**Stage-1: HSIP-guided Scale and Root Refinement.** This stage explicitly optimizes the global scene scale and root positions using geometric and crowd-structure priors. The residual scale parameter $\alpha$ is optimized jointly with root translation and orientation $(\boldsymbol{\tau}, \boldsymbol{\gamma})$. HSIP-based feasible-region constraints, including $\mathcal{E}_{\text{HSIP}}^{(proj)}$, $\mathcal{E}_{\text{HSIP}}^{(xz)}$, and $\mathcal{E}_{\text{HSIP}}^{(y)}$, are the primary drivers that align the crowd distribution with the scene geometry and resolve global scale ambiguity. Based on the HSIP anchoring, root positions are further refined and stabilized by the crowd-level structural regularization $\mathcal{E}_{\text{crowd}}^{(t)}$ together with temporal smoothness terms on root translation and rotation, $\mathcal{E}_{\boldsymbol{\tau}}^{(t)}$ and $\mathcal{E}_{\boldsymbol{\gamma}}^{(t)}$, yielding stable and geometrically plausible root trajectories.

**Stage-2: Pixel-consistent Pose Refinement.** With the global scale and root trajectories stabilized, this stage refines per-frame articulated body poses using pose-level priors and image evidence. The pose latent code $\mathbf{z}_\theta$ is optimized under the VPoser prior (Pavlakos et al., 2019) $\mathcal{E}_{\text{prior}}^{\theta}$ to ensure anatomically plausible configurations. Pixel-level joint consistency is enforced through the reprojection loss $\mathcal{E}_{\text{proj}}$, while temporal pose smoothness is encouraged by $\mathcal{E}_{\boldsymbol{\theta}}^{(t)}$ to maintain stable articulation over time.

**Stage-3: Group-guided Motion Prior Refinement.** This stage recovers temporally coherent motion dynamics by optimizing the motion latent code $\mathbf{z}_\xi$ under the learned motion prior $\mathcal{E}_{\text{prior}}^{\xi}$, following the motion prior formulation in DyCrowd (Wen et al., 2025). Contact-aware motion regularization is imposed via the contact velocity term $\mathcal{E}_{\text{contact}}^{(v)}$ to suppress sliding artifacts, and temporal continuity across adjacent motion segments is enforced by the segment con-

nection loss $\mathcal{E}_{\text{connect}}$. In addition, group-guided motion consistency encoded by $\mathcal{E}_{\text{AMC}}$ propagates reliable dynamics across individuals, improving robustness in crowded and occluded scenarios.

### 3.4. HSIP-based Crowd and Scene Alignment

To ensure scale-consistent and physically plausible human–scene relationships, we explicitly introduce crowd–scene interaction constraints during joint optimization. We leverage the Human–Scene Interaction Proxy (HSIP) defined in Sec. 3.2.2 to align the crowd distribution with the reconstructed scene geometry. A residual global scene scale $\alpha \in \mathbb{R}^+$ is treated as an optimizable variable and initialized as 1. The HSIP points $\mathbf{s}_{n,t}^*$ and their support ranges $r_{n,t}^{\text{HSIP}}$ are constructed in the canonical frame scaled by $\alpha_0$ (Eq. (2)), while $\alpha$ further refines crowd–scene alignment during optimization.

By constraining human root translations using the dynamically scaled HSIP, we jointly stabilize the global scene scale and enforce valid placement of individuals on complex terrain. We decompose this consistency into horizontal feasible-region alignment, vertical terrain adaptation, and projection consistency. In addition, once the global scale becomes stable during optimization, we apply a direct root position update by setting the root translation to the HSIP-corresponding reference position along the viewing ray, i.e., $\boldsymbol{\tau}_{n,t} \leftarrow \mathbf{r}_{n,t}^*$.

**Horizontal Feasible Region Alignment.** The HSIP provides a scene-grounded proxy location $\mathbf{s}_{n,t}^*$ together with a local support range $r_{n,t}^{\text{HSIP}}$. We constrain the horizontal root translation $(\boldsymbol{\tau}_{n,t})_{xz}$ to remain within the scaled feasible region centered at $\alpha(\mathbf{s}_{n,t}^*)_{xz}$. A hinge loss penalizes deviations when the person drifts outside this region:

$$\mathcal{E}_{\text{HSIP}}^{(xz)} = \lambda_{xz} \sum_{(n,t) \in \Omega} \left[ \left\| (\boldsymbol{\tau}_{n,t})_{xz} - \alpha\,(\mathbf{s}_{n,t}^*)_{xz} \right\|_2 - \alpha\,r_{n,t}^{\text{HSIP}} \right]_+^2 . \tag{17}$$

**Vertical Terrain Adaptation.** The HSIP point $\mathbf{s}_{n,t}^*$ provides a terrain height reference. We constrain the lowest vertex of the human mesh, $\mathbf{v}_{n,t}^\downarrow$, to match the scaled HSIP height:

$$\mathcal{E}_{\text{HSIP}}^{(y)} = \lambda_y \sum_{(n,t) \in \Omega} \left( (\mathbf{v}_{n,t}^\downarrow)_y - \alpha\,(\mathbf{s}_{n,t}^*)_y \right)^2 . \tag{18}$$

**Projection Consistency.** To preserve image-space alignment, we introduce a projection consistency constraint. The effective anchor is defined as the projection of the root translation onto the horizontal plane:

$$\mathbf{a}_{n,t}^{\text{proj}} = \left( x(\boldsymbol{\tau}_{n,t}),\ y(\mathbf{v}_{n,t}^\downarrow),\ z(\boldsymbol{\tau}_{n,t}) \right). \tag{19}$$

We enforce consistency between its image projection and that of the probe anchor:

$$\mathcal{E}_{\text{HSIP}}^{(proj)} = \lambda_{\text{proj}} \sum_{(n,t)\in\Omega} \left\| \Pi_t\left(\mathbf{a}_{n,t}^{\text{proj}}\right) - \Pi_t\left(\mathbf{a}_{n,t}'\right) \right\|_2^2, \quad (20)$$

where $\Pi_t(\cdot)$ denotes the perspective projection under camera parameters $\bar{\mathcal{C}}_t$.

### 3.5. Crowd Structural Coherence Regularization

Per-person temporal smoothness alone becomes unreliable in dense crowds due to frequent occlusions and depth ambiguity. We therefore introduce *Crowd Structural Coherence Regularization* (CSCR), which leverages HSIP-based spatial priors to impose soft temporal consistency on pairwise relative displacements within local crowd neighborhoods.

**Dynamic Interaction Graph.** At each frame $t$, we construct a directed interaction graph $\mathcal{G}_t = (\mathcal{V}_t, \mathcal{E}_t)$, where nodes correspond to visible individuals. Neighborhood relations are determined in the horizontal plane using the HSIP points $(\mathbf{s}_{n,t}^*)_{xz}$ (Sec. 3.2.2), and edges $(n,m) \in \mathcal{E}_t$ connect spatial neighbors $m \in \mathcal{N}_n^t$ with interaction weights $w_{nm}^t$ based on spatial proximity. A temporal stride $\Delta$ is used to evaluate structural consistency over time.

**Crowd Structural Coherence Loss.** Let $\mathbf{u}_{nm}^t = \boldsymbol{\tau}_{n,t} - \boldsymbol{\tau}_{m,t}$ denote the 3D relative displacement between neighboring individuals. The crowd structural coherence energy is defined as

$$\mathcal{E}_{\text{crowd}}^{(t)} = \lambda_{\text{crowd}} \sum_{t>\Delta} \sum_{(n,m)\in\mathcal{E}_t} \tilde{w}_{nm}^t \Bigg[ \rho\big(\|\mathbf{u}_{nm}^t - \mathbf{u}_{nm}^{t-\Delta}\|_2\big)$$
$$+ \lambda_{\text{dir}}\, \rho\left(1 - \frac{\mathbf{u}_{nm}^t \cdot \mathbf{u}_{nm}^{t-\Delta}}{\|\mathbf{u}_{nm}^t\|_2 \|\mathbf{u}_{nm}^{t-\Delta}\|_2 + \varepsilon}\right) \Bigg],$$
$$(21)$$

where $\tilde{w}_{nm}^t$ is the effective interaction weight, $\rho(\cdot)$ denotes the Charbonnier penalty, $\lambda_{\text{dir}}$ balances direction-aware consistency, and $\varepsilon$ ensures numerical stability. The loss is evaluated only for pairs validly tracked at both $t$ and $t - \Delta$.

## 4. Experiments

### 4.1. Datasets and Evaluation Metrics

**Datasets.** We evaluate Crowd4D on VirtualCrowd (Wen et al., 2025), a synthetic benchmark with complex terrain and dense crowds (60–200 people per scene). Since Crowd4D is optimization-based and requires no training, VirtualCrowd is used only for quantitative evaluation. We additionally report qualitative results on the real-world PANDA dataset (Wang et al., 2020), which provides gigapixel-level videos but no 3D pose annotations.

**Evaluation Metrics.** We evaluate global crowd-level consistency using distance-based metrics, including PPDS and its Procrustes-aligned variant (PA-PPDS) (Wen et al., 2023), as well as the percentage of correct ordinal depth (PCOD) (Zhen et al., 2020). For pose accuracy, we report MPJPE and PA-MPJPE. To assess global-coordinate pose sequences, we additionally report WA-MPJPE and W-MPJPE following (Ye et al., 2023). Motion smoothness is measured by the acceleration error (ACCEL).

### 4.2. Implementation Details

We implement our method in PyTorch (Paszke et al., 2019) and optimize all models using the Adam optimizer. The optimization is carried out in three successive stages (stage-1, stage-2, and stage-3), with 150, 150 and 300 iterations, respectively. The learning rates are set to 0.01, 0.01, and 0.02 for the three stages. All experiments are conducted on a workstation equipped with an NVIDIA 4090D GPU and 128 GB of system memory. Optimizing a large-scale scene containing 100 individuals over 200 frames takes approximately 4 hours. Reducing the optimization budget to 100 iterations per stage enables approximately hour-level reconstruction while providing a practical trade-off between efficiency and reconstruction quality.

### 4.3. Comparison

Following DyCrowd (Wen et al., 2025), we compare our method with state-of-the-art crowd reconstruction approaches on the VirtualCrowd dataset under two tracking settings: *Unified Tracking-by-Detection*, which uses detected trajectories and therefore retains practical identity association errors, and *Ground-truth Object Tracking*, which uses ground-truth identity associations to isolate reconstruction quality from tracking errors (Table 2).

For large-scale dynamic crowd reconstruction, DyCrowd serves as the primary baseline, together with image-based methods Crowd3D and GroupRec. Under the unified tracking-by-detection protocol, our method consistently outperforms DyCrowd on global and crowd-level metrics, achieving a +5.8 improvement in PPDS and a smaller PPDS–PA-PPDS gap, which indicates better global crowd consistency. We also reduce MPJPE by 7.9 mm without relying on post-hoc alignment. Under the ground-truth object tracking setting, our method achieves the best global consistency, with PPDS reaching 91.43 (+6.8 over DyCrowd) and MPJPE reduced by 9.5 mm.

Our method further demonstrates stronger robustness to occlusions than DyCrowd, as shown in Table 3 and Fig. 4, particularly under severe occlusions and depth ambiguity. Moreover, our reconstructions exhibit more temporally coherent and physically plausible motion, and results on the PANDA dataset under flat and non-flat terrain settings show

| Method | | PPDS↑ | PA-PPDS↑ | PCOD↑ | MPJPE↓ | PA-MPJPE↓ | WA-MPJPE↓ | W-MPJPE↓ | ACCEL↓ |
|---|---|---|---|---|---|---|---|---|---|
| **Unified Tracking-by-Detection** | | | | | | | | | |
| Crowd3D (Wen et al., 2023) | CVPR'23 | 82.61 | 87.98 | 91.11 | 122.99 | 73.70 | - | - | - |
| GroupRec (Huang et al., 2023) | ICCV'23 | 74.25 | 75.04 | 86.22 | 89.04 | 58.98 | 82.20 | 94.34 | 165.50 |
| DyCrowd (Wen et al., 2025) | TPAMI'25 | 83.21 | 89.10 | 92.20 | 69.74 | 48.57 | 68.99 | 83.39 | **15.72** |
| Ours | | **89.04** | **89.46** | **92.92** | **61.83** | **45.32** | **65.77** | **74.15** | 16.10 |
| **Ground-truth Object Tracking** | | | | | | | | | |
| SLAHMR-Large*(Ye et al., 2023) | CVPR'23 | 84.41 | 87.95 | 90.34 | 106.35 | 69.41 | 90.10 | 108.40 | **12.25** |
| DyCrowd (Wen et al., 2025) | TPAMI'25 | 84.66 | 91.23 | 95.38 | 68.81 | 45.34 | 65.91 | 80.34 | 15.53 |
| Ours | | **91.43** | **92.38** | **95.57** | **59.35** | **44.36** | **63.98** | **73.32** | 12.93 |

*Table 2.* Quantitative Comparison on VirtualCrowd Dataset. while "-" means unavailable results, "*" indicates methods modified for large-scene videos. The best and second-best results are highlighted in **bold** and underline, respectively.

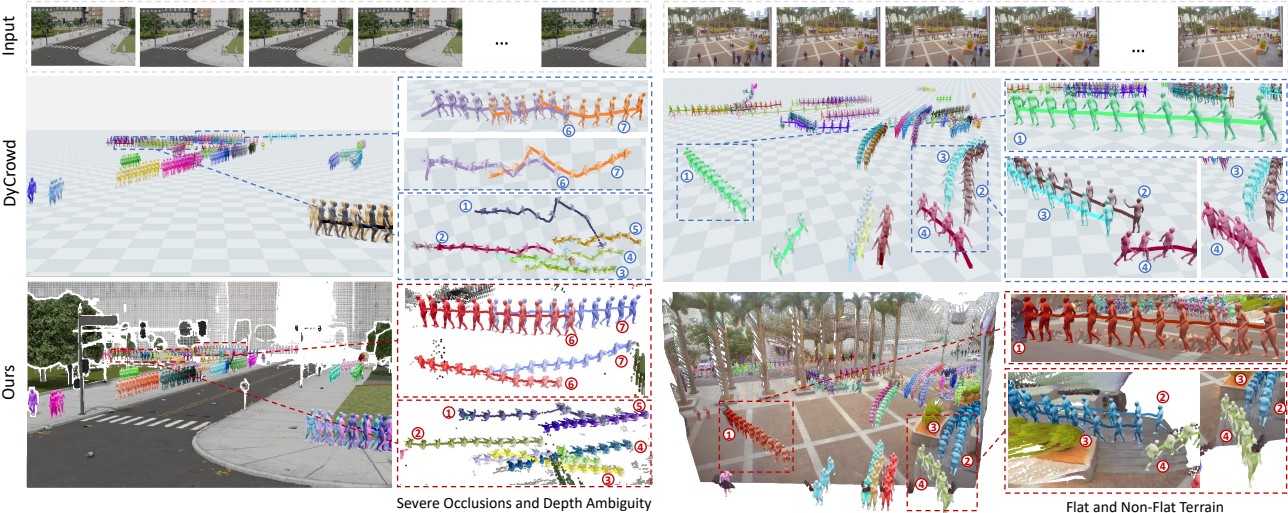

*Figure 4.* Qualitative comparison between our method and DyCrowd on the VirtualCrowd and PANDA datasets. Identical indices denote the same individual. The highlighted regions illustrate results under Flat and Non-Flat Terrain as well as Severe Occlusions and Depth Ambiguity.

| Method | MPJPE↓ | PA-MPJPE↓ |
|---|---|---|
| DyCrowd (Wen et al., 2025) | 66.09 | 56.21 |
| Ours | 64.19 | 54.81 |

*Table 3.* Quantitative Comparison on Occluded Instances of the VirtualCrowd Dataset.

| Method | PPDS↑ | PA-PPDS↑ | PCOD↑ | MPJPE↓ | ACCEL↓ |
|---|---|---|---|---|---|
| VideoMimic | 80.03 | 80.09 | 91.34 | 62.49 | 23.80 |
| Ours | **91.43** | **92.38** | **96.57** | **59.35** | **12.93** |

*Table 4.* Quantitative comparison with the small-scene method VideoMimic (Allshire et al., 2025) under the ground-truth object tracking protocol.

more reasonable alignment with the scene geometry.

We further adapt the small-scene method VideoMimic (Allshire et al., 2025) to large-scale scenarios using the same reconstructed cameras, scene geometry, and evaluation protocol as our method. Since it relies primarily on iterative depth alignment between humans and the scene, VideoMimic lacks constraints for maintaining globally coherent crowd layouts at scale, resulting in a substantially lower PPDS of 80.03, compared with 91.43 achieved by our method in

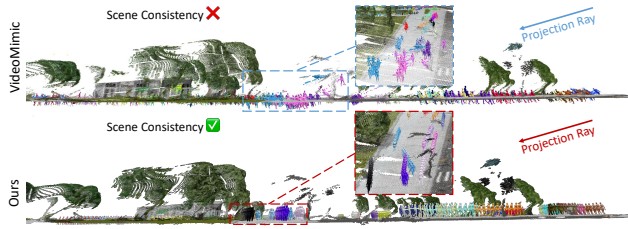

*Figure 5.* Qualitative comparison between our method and the small-scene method VideoMimic (Allshire et al., 2025) from a side-view perspective on the VirtualCrowd dataset.

Table 4. As shown in Fig. 5, this leads to accumulated depth drift and implausible crowd configurations.

### 4.4. Ablations

The Multi-stage Optimization Strategy ablation is evaluated on all test videos of VirtualCrowd. For the remaining ablations, we use two selected videos from Scene 1 with distinct viewpoints (a higher and a lower view) to facilitate controlled analysis.

**Multi-stage Optimization Strategy.** As shown in Table 5,

| Stages | PPDS↑ | PA-PPDS↑ | PCOD↑ | MPJPE↓ | ACCEL↓ |
|---|---|---|---|---|---|
| Stage-1 only | 88.76 | 89.19 | 92.67 | 76.23 | 34.85 |
| Stage-1 + Stage-2 | 88.88 | 89.34 | 92.89 | 67.28 | 62.92 |
| Stage-1 + Stage-2 + Stage-3 | **89.04** | **89.46** | **92.92** | **61.83** | **16.10** |

*Table 5.* Ablation study of the multi-stage optimization strategy.

| Method | PPDS↑ | PA-PPDS↑ | PCOD↑ | MPJPE↓ | ACCEL↓ |
|---|---|---|---|---|---|
| w/o joint processing | 67.92 | 68.40 | 77.38 | 154.08 | 41.04 |
| w/ joint distance | 79.26 | 79.22 | 89.50 | 62.89 | 25.58 |
| w/ direct projection anchor | 84.81 | 87.95 | 92.22 | 65.73 | 24.80 |
| Ours (w/ HSIP Alignment) | **90.71** | **91.22** | **93.57** | **61.76** | **16.66** |

*Table 6.* Ablation study on human-scene joint processing strategies.

| Method | PPDS↑ | PA-PPDS↑ | PCOD↑ | MPJPE↓ | ACCEL↓ |
|---|---|---|---|---|---|
| w/o $\mathcal{E}_{\text{crowd}}^{(t)}$ | 90.57 | 91.04 | 93.44 | 66.24 | 16.79 |
| Ours | **90.71** | **91.22** | **93.57** | **61.76** | **16.66** |

*Table 7.* Ablation study of crowd structural coherence regularization.

our stage-wise optimization steadily improves performance. Stage-1 provides a strong initialization with high global consistency (PPDS/PCOD = 88.76/92.67) but relatively large pose error (MPJPE = 76.23). Stage-2 mainly focuses on pose refinement, reducing MPJPE to 67.28 while keeping global metrics nearly unchanged (PPDS/PCOD = 88.88/92.89). Finally, Stage-3 further refines the full sequence, improving both pose accuracy and motion smoothness (MPJPE = 61.83, ACCEL = 16.10).

**Human-Scene Joint Processing.** Table 6 compares different strategies for coupling humans with reconstructed scene geometry. Removing joint processing leads to severe scale drift and inconsistent crowd layouts. Joint distance constraints partially reduce drift but remain sensitive to noisy scene points. Projection-based anchors (Crowd3D (Wen et al., 2023)) improve global metrics in large scenes but lack a temporally stable proxy, making them prone to amplified 3D errors under low viewing angles. In contrast, our HSIP-based crowd–scene alignment (Sec. 3.4) achieves the best global consistency (PPDS, PCOD) and motion smoothness (ACCEL).

**Crowd Structural Coherence Regularization.**

We ablate the proposed crowd structural coherence term $\mathcal{E}_{\text{crowd}}^{(t)}$, which enforces temporal consistency of pairwise relative displacements within HSIP-induced local neighborhoods. As shown in Table 7, removing $\mathcal{E}_{\text{crowd}}^{(t)}$ slightly degrades global crowd consistency (PPDS, PA-PPDS, PCOD), but leads to a clear drop in pose accuracy, with MPJPE increasing from 61.76 to 66.24. This indicates that CSCR plays an important role in stabilizing local crowd structure over time, especially under occlusions, and contributes to more temporally coherent and physically plausible motion.

## 5. Conclusion

We proposed Crowd4D for scene-consistent monocular 4D crowd reconstruction in large, complex-terrain scenes. By jointly optimizing crowd motion and a residual global scale, and introducing HSIP built on SIPC&SIS for scene-aware and temporally stable crowd–scene anchoring together with CSCR for neighborhood-level temporal coherence, Crowd4D produces globally consistent and temporally coherent reconstructions across diverse large-scale scenes. Nevertheless, Crowd4D remains an offline optimization framework with non-negligible computational cost, and its performance depends on the quality of monocular scene reconstruction, human initialization, and tracking, especially under severe occlusions. In addition, the absence of pose-level 3D annotations in real-world large-scale crowd datasets limits quantitative evaluation in real scenes. Future work will investigate more efficient feed-forward approximations and pseudo-label-based learning.

## Acknowledgments

This work was supported in part by National Key R&D Program of China (2023YFC3082100) and Science Fund for Distinguished Young Scholars of Tianjin (No.22JCJQJC00040).

## Impact Statement

This paper presents work whose goal is to advance the field of Machine Learning. On the positive side, the proposed research can be useful for urban design and management, public safety, intelligent transportation, crowd simulation, etc. Like many other works involving human reconstruction from visual data, the work may also lead to concerns regarding privacy. Our method does not reconstruct facial details, which are often crucial for individual recognition. The technology should only be used in compliance with local regulations/laws, including camera placements and transparent communication.

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

# A. All Energy Functions.

This appendix summarizes the energy terms used in the three-stage optimization strategy described in Sec. 3. The correspondence between each energy term and its weight is provided in the main text (Table 1). Note that the HSIP-based crowd–scene alignment energies $\mathcal{E}_{\text{HSIP}}^{(xz)}$, $\mathcal{E}_{\text{HSIP}}^{(y)}$, and $\mathcal{E}_{\text{HSIP}}^{(proj)}$ are introduced in the main text (Sec. 3.4), and the CSCR term $\mathcal{E}_{\text{crowd}}^{(t)}$ is defined in Sec. 3.5; thus, they are not repeated here.

**Observation Consistency Constraint.** To align the reconstructed motion with visual observations, we employ a scale-normalized 2D reprojection loss. Using the camera parameters $\bar{\mathcal{C}}_t$, we project the world-space joints $\mathbf{J}_{n,t,j}$ onto the image plane via the projection operator $\pi(\cdot)$. To balance gradients across individuals at varying depths, we normalize the residuals by a scale factor $\bar{A}_{n,t} = \sqrt{w_{n,t} h_{n,t}}$, derived from the bounding box. The energy is computed over the set of valid keypoints $\Omega_{\text{reproj}}$:

$$\mathcal{E}_{\text{proj}} = \lambda_{\mathbf{J}^{2D}} \sum_{(n,t,j)\in\Omega_{\text{reproj}}} \left\| \frac{\pi(\mathbf{J}_{n,t,j}; \bar{\mathcal{C}}_t) - \bar{\mathbf{J}}_{n,t,j}^{2D}}{\bar{A}_{n,t}} \right\|_2^2 . \tag{22}$$

**Per-individual Temporal Smoothness.** To reduce temporal jitter and improve motion stability, we impose per-individual temporal smoothness constraints on global translation, root orientation, and articulated body pose. Concretely, we penalize inter-frame translation velocities ($\mathcal{E}_{\boldsymbol{\tau}}^{(t)}$) and relative angular velocities in the Lie algebra for both the root orientation and joint rotations ($\mathcal{E}_{\boldsymbol{\gamma}}^{(t)}$ and $\mathcal{E}_{\boldsymbol{\theta}}^{(t)}$).

For global translation $\boldsymbol{\tau}_{n,t}$, we penalize inter-frame velocities:

$$\mathcal{E}_{\boldsymbol{\tau}}^{(t)} = \lambda_{\tau} \sum_{(n,t)\in\Omega_{\text{adj}}} \|\boldsymbol{\tau}_{n,t+1} - \boldsymbol{\tau}_{n,t}\|_2^2 . \tag{23}$$

We apply a unified regularization strategy for both root orientation $\boldsymbol{\gamma}$ and body pose $\boldsymbol{\theta}$. For body pose, we average the loss over $J$ joints to keep gradients balanced across joints. Let $\mathbf{R}(\mathbf{v})$ denote the rotation matrix corresponding to an axis-angle vector $\mathbf{v}$. The losses are formulated as:

$$\mathcal{E}_{\boldsymbol{\gamma}}^{(t)} = \lambda_{\gamma} \sum_{(n,t)\in\Omega_{\text{adj}}} \left\| \log\left( \mathbf{R}(\boldsymbol{\gamma}_{n,t})^{\top} \mathbf{R}(\boldsymbol{\gamma}_{n,t+1}) \right) \right\|_2^2 \tag{24}$$

$$\mathcal{E}_{\boldsymbol{\theta}}^{(t)} = \lambda_{\theta} \sum_{(n,t)\in\Omega_{\text{adj}}} \frac{1}{J} \sum_{j=1}^{J} \left\| \log\left( \mathbf{R}(\boldsymbol{\theta}_{n,t}^{(j)})^{\top} \mathbf{R}(\boldsymbol{\theta}_{n,t+1}^{(j)}) \right) \right\|_2^2 ,$$

**Shape Prior.** To prevent unnatural body shapes, we impose an isotropic Gaussian prior on the SMPL shape parameters:

$$\mathcal{E}_{\boldsymbol{\beta}} = \lambda_{\beta} \sum_{(n,t)\in\Omega} \|\boldsymbol{\beta}_{n,t}\|_2^2 . \tag{25}$$

**Pose Prior.** We regularize the articulated pose with a Gaussian prior in the VPoser latent space. Let $\tilde{\mathbf{z}}_{n,t}^{\theta} \in \mathbb{R}^{32}$ denote the optimizable VPoser (Pavlakos et al., 2019) latent code corresponding to the SMPL pose at frame $t$. We define

$$\mathcal{E}_{\text{prior}}^{\theta} = \lambda_{\theta}^{\text{prior}} \sum_{(n,t)\in\Omega} \|\tilde{\mathbf{z}}_{n,t}^{\theta}\|_2^2 , \tag{26}$$

where the equivalence holds up to an additive constant.

**Motion Prior.** We follow DyCrowd (Wen et al., 2025) and introduce motion-level priors and group-level consistency to stabilize long sequences under occlusions. For each person $n$, we split the motion into $S_n$ overlapping temporal segments and optimize a motion latent code $\mathbf{z}_{n,s}^{\xi}$ for each segment. Assuming a standard normal prior, we penalize deviations in the latent space:

$$\mathcal{E}_{\text{prior}}^{\xi} = \lambda_{\text{mp}} \sum_{n} \sum_{s=1}^{S_n} \|\mathbf{z}_{n,s}^{\xi}\|_2^2 . \tag{27}$$

**Contact Stationarity.** Let $c_{n,t,j} \in [0,1]$ denote the predicted contact confidence of joint $j$ for person $n$ at time $t$, and let $\Delta_t \mathbf{J}_{n,t,j} = \mathbf{J}_{n,t+1,j} - \mathbf{J}_{n,t,j}$ denote the inter-frame joint displacement in world space. We encourage contacting joints to remain stationary:

$$\mathcal{E}_{\text{contact}}^{(v)} = \lambda_{\text{contact}} \sum_n \sum_t \sum_j c_{n,t,j} \left\| \Delta_t \mathbf{J}_{n,t,j} \right\|_2^2. \tag{28}$$

**Segment Connection.** To ensure continuity across adjacent segments, we penalize the joint discrepancy between the last frame $s_l$ of a segment and the first frame $s_1^+$ of its subsequent segment:

$$\mathcal{E}_{\text{connect}} = \lambda_{\text{con}} \sum_{n,s} \left\| \mathbf{J}_{n,s_1^+} - \mathbf{J}_{n,s_l} \right\|_2^2. \tag{29}$$

**Group-guided AMC.** To recover occluded or unreliable motions in video sequences, we adopt the Asynchronous Motion Consistency (AMC) loss from DyCrowd (Wen et al., 2025). Within each segment, unreliable individuals are aligned to confident group references using Soft-DTW:

$$\mathcal{E}_{\text{AMC}} = \lambda_{\text{amc}} \sum_s \sum_{n \in \mathcal{U}_s} w_{n,s} \, \text{sDTW}(\mathbf{P}_{n,s}, \mathbf{P}_{\text{ref},s}). \tag{30}$$

Here, $\mathbf{P}_{n,s}$ denotes the pose sequence of person $n$ in segment $s$, and $\mathbf{P}_{\text{ref},s}$ is selected from the most confident group member or a canonical template when unavailable.

