# OpenReview forum: "Crowd4D: Scene-Aware Monocular 4D Crowd Reconstruction"
_ICML.cc/2026/Conference — ICML 2026 regular_

### Official Review · Reviewer_C3iu · 2026-03-10

**Soundness:** 3
**Presentation:** 3
**Significance:** 2
**Originality:** 2
**Overall Recommendation:** 3
**Confidence:** 4

**Summary:**

The paper addresses the problem of reconstructing scene-consistent 4D crowd motion from monocular RGB video in large-scale, complex environments. Existing methods often suffer from depth ambiguity and scale drift because they rely on simplified single-plane ground assumptions. The authors propose Crowd4D, a framework that jointly optimizes crowd motion and scene scale. Key components include:

A representation derived from Scene Interaction Point Clouds and Surfaces (SIPC & SIS) that anchors human placement to complex terrain.
A group-level constraint that uses local neighborhood topology to maintain temporal stability, especially during occlusions.
A three-stage strategy progressively refining scale, root positions, articulated poses, and motion priors.

**Compliance With Llm Reviewing Policy:**

Affirmed.

**Key Questions For Authors:**

In Table 7, removing the CSCR term leads to a significant increase in MPJPE (from 61.76 to 66.24) but only a slight change in global metrics like PPDS. Can the authors explain why a group-level spatial constraint has such a pronounced effect on local articulated pose accuracy?

 The authors estimate a global scale factor α by aligning human depth with scene geometry. How does the system handle scenarios where the initial scene reconstruction is significantly distorted or lacks enough geometric detail for a reliable SIS?

Given the 4-hour optimization time, have the authors explored any distillation or feed-forward approximations to make the framework more practical?

**Limitations:**

yes

**Strengths And Weaknesses:**

Strengths:
Unlike previous large-scale methods (e.g., Crowd3D, DyCrowd) that assume a flat ground plane, Crowd4D explicitly models complex geometry using the SIPC and SIS.

The introduction of CSCR leverages the idea that while individual motion is dynamic, the local structural topology of a crowd is often coherent. This effectively reduces jitter in dense, occluded scenarios.

 The method outperforms state-of-the-art baselines like DyCrowd on the VirtualCrowd dataset. Qualitative results demonstrate effectiveness across diverse settings, from gigapixel surveillance to drone-captured videos with dynamic camera motion


Weaknesses:

The optimization process is extremely slow. This limits its applicability for real-time urban management or public safety tasks mentioned in the introduction.

The pipeline is highly modular and relies on several pre-existing, heavy-duty tools for initialization (e.g. for scene reconstruction, GeoCalib for gravity, and various trackers/pose estimators). The "scene-aware" nature is largely dependent on the quality of these initial reconstructions.

While the HSIP is a novel anchoring mechanism, much of the Stage-3 optimization, including the motion prior, contact-aware regularization, and AMC loss, is directly adopted from DyCrowd.

Quantitative metrics are primarily reported on the synthetic VirtualCrowd dataset. Evaluation on the real-world PANDA dataset is limited to qualitative observations due to a lack of 3D annotations. More evaluation is needed

---

> ### Author Rebuttal · Authors · 2026-03-31
>
> We thank Reviewer C3iu for the positive evaluation. Below we address the raised concerns.
>
> **W1. The optimization process is extremely slow**
>
> The main bottleneck is the iterative three-stage joint optimization (150/150/300 iterations), where Stage-3 accounts for approximately two-thirds of the total runtime. A lighter mode is feasible: Table 5 shows Stage-1 already provides strong global consistency (PPDS=88.76). By reducing iterations or running only the first two stages, runtime can be reduced to under 1 hour with moderate accuracy trade-off. Additionally, multi-GPU parallelization could further reduce runtime. For context, TRAM (a single-person regression method without scene reconstruction) requires ~1 minute per person for 200 frames; scaling to 100 people would need at least 1.5 hours. Direct regression for crowds remains challenging due to lack of 3D training data. Our key advantage is generating high-quality pseudo ground-truth to alleviate this data scarcity. We will provide lightweight configurations for different needs and detailed per-stage runtime analysis in the revision.
>
> **W2. The pipeline is highly modular and relies on several pre-existing tools**
>
> This modular design is common practice—VideoMimic similarly rely on external tools for camera and scene geometry. We incorporate scene priors and joint optimization to ensure consistency between crowds, images, and geometry. While scene-aware reconstruction depends on initial geometry quality, π³ provides stable reconstruction in our experiments. We will include sensitivity analysis to initialization tools in the revision.
>
> **W3. Much of the Stage-3 optimization is directly adopted from DyCrowd**
>
> While certain energy terms are inspired by DyCrowd, our Stage-3 operates within a fundamentally different framework. Rather than planar assumptions, we introduce HSIP for complex non-planar geometries with dedicated terms for uneven terrain. Our core innovations (HSIP and CSCR) significantly improve crowd position estimation in early stages. Detection noise causes large spatial errors that severely affect motion prior optimization—evident in DyCrowd (Figure 4). By optimizing scale and spatial alignment first, we provide reliable initialization for Stage-3, enabling it to operate in a cleaner solution space. The gains stem from fundamentally different optimization design.
>
> **W4. Quantitative metrics are primarily reported on the synthetic VirtualCrowd dataset**
>
> PANDA provides bounding box and trajectory annotations but no pose data (neither 2D nor 3D). A fundamental challenge is the lack of large-scale 3D crowd datasets with pose annotations. Following DyCrowd, we conduct quantitative evaluation on VirtualCrowd with controlled 3D annotations while providing qualitative results on PANDA for real-world validation. Notably, our optimization-based approach can generate 3D pseudo ground-truth from real videos, potentially addressing the critical 3D crowd data scarcity.
>
> **Q1. Why does CSCR have such a pronounced effect on local articulated pose accuracy?**
>
> This stems from metric sensitivities. PPDS measures global distribution consistency of crowds (hundreds of people), making it insensitive to local corrections. MPJPE evaluates individual joint accuracy, making it sensitive to local errors. CSCR operates at the local neighborhood level, strengthening displacement consistency between adjacent individuals. In crowds, inaccurate root positions propagate to pose estimation, causing amplified MPJPE errors. CSCR optimizes local spatial relationships, bringing misaligned individuals to reasonable configurations, significantly reducing MPJPE while marginally affecting PPDS.
>
> **Q2. How does the system handle scenarios where the initial scene reconstruction is significantly distorted?**
>
> In typical crowd scenarios, open environments and long sequences provide sufficient geometric cues for reliable reconstruction. We acknowledge severely distorted geometry remains challenging, even in small-scale settings where robust monocular reconstruction is not fully solved. Our method alleviates this through: HSIP-based anchoring utilizing temporally aggregated scene information and feasible regions, providing robustness to local noise; CSCR providing constraints through relative spatial relationships, partially compensating for imperfect geometry via crowd distribution characteristics. We will extend failure case discussions in the revision.
>
> **Q3. Have the authors explored any distillation or feed-forward approximations?**
>
> We agree these are important for practical efficiency. Our primary focus is addressing the lack of reliable large-scale 4D crowd supervision data. Obtaining accurate 3D crowd motion data is challenging due to occlusion, scale variations, and lack of motion capture. Our approach generates high-quality pseudo ground-truth for subsequent learning—a crucial step toward enabling future feed-forward model training or distillation.

---

### Official Review · Reviewer_4KCy · 2026-03-11

**Soundness:** 3
**Presentation:** 3
**Significance:** 3
**Originality:** 3
**Overall Recommendation:** 4
**Confidence:** 3

**Summary:**

This work focuses on the task of scene-aware 4D crowd reconstruction. Specifically, this work proposes the Crowd4D framework for the scene-aware 4D crowd reconstruction. Crowd4D introduces Human–Scene Interaction Proxy (HSIP), which is built upon Scene Interaction Point Clouds and a Scene Interaction Surface (SIPC&SIS). Furthermore, Crowd4D introduces Crowd Structural Coherence Regularization (CSCR). Experimental results show that the proposed Crowd4D achieves better performance compared to previous baselines.

**Compliance With Llm Reviewing Policy:**

Affirmed.

**Final Justification:**

My concerns have been adequately addressed. The authors have provided further results. I have no further questions, and I keep my score of 4.

**Key Questions For Authors:**

* I am not very familiar with the specific task of monocular 4D crowd reconstruction. Are the methods in Table 1 the mainstream methods in this field? I feel that the methods are slightly outdated.
* I noticed that some metrics in the ablation experiments in Tables 5 and 7 showed slight changes. Although the article has provided a brief analysis, I wonder if a more detailed explanation and analysis of this phenomenon is needed?

**Limitations:**

The Impact Statement is involved in this manuscript, but I did not find the declaration of limitations of this work.

**Strengths And Weaknesses:**

**Strength**

* The overall writing of this work is good, and the readers can understand the proposed method.
* I think the focused task of monocular 4D crowd reconstruction is meaningful for practice.
* The experimental results basically demonstrate the effectiveness of this work.

**Weakness**

* The qualitative results of ablations are required. I did not find them in the manuscript.
* This work presents the computational cost of the proposed method. I think the computational cost comparisons with other baselines are also required.
* The Impact Statement is involved in this manuscript, but I did not find the declaration of limitations of this work. Please supplement it.

---

> ### Author Rebuttal · Authors · 2026-03-31
>
> We thank Reviewer 4KCy for the positive evaluation and for recognizing the writing quality and practical significance of our work. Below we address the raised concerns and questions.
>
> **W1. The qualitative results of ablations are required**
>
> Thank you for this valuable suggestion. We will supplement qualitative visualizations for the ablation studies in the revised version, particularly demonstrating the visual differences contributed by HSIP and CSCR respectively. These visualizations will help readers better understand how each component affects the reconstruction quality.
>
> From our qualitative observations: (1) without HSIP, reconstructed individuals exhibit noticeable vertical drift and ground penetration artifacts, particularly on sloped or uneven terrain; (2) without CSCR, adjacent individuals under heavy occlusion show spatially inconsistent jitter across consecutive frames, as inter-person structural constraints are absent. We will include these visualizations in the revised supplementary material.
>
> **W2. The computational cost comparisons with other baselines are also required**
>
> Thank you for this suggestion. The reported runtime of Crowd4D is measured under optimal settings to ensure the best accuracy. The overall efficiency depends on the number of optimization iterations and the computational complexity of energy functions. Under identical conditions, Crowd4D achieves lower runtime than DyCrowd while ensuring better accuracy. Furthermore, as shown in Table 5, the first two stages already achieve strong global consistency. For applications prioritizing efficiency, users can run only the first two stages or reduce iteration steps to achieve faster processing. We provide flexible configuration options for different application scenarios. For pseudo ground-truth level results, we recommend using the optimal settings. We will include detailed comparisons and discussions in the revised version.
>
> **W3. The declaration of limitations of this work is missing**
>
> Thank you for pointing this out. While the manuscript includes an Impact Statement, we acknowledge that an explicit limitations declaration is indeed missing. We will add a dedicated limitations section in the revised version, covering: (1) dependency on scene reconstruction quality, (2) computational efficiency constraints, (3) failure modes in extreme scenarios, and (4) current limitations in real-world quantitative evaluation due to lack of 3D annotations.
>
> **Q1. Are the methods in Table 1 the mainstream methods in this field? Are they outdated?**
>
> The methods in Table 1 represent the mainstream approaches in this field. Large-scale dynamic crowd reconstruction is an emerging research area with limited existing work. DyCrowd (TPAMI'25) is the first method to address dynamic large-scene crowd reconstruction and serves as the primary baseline in our evaluation (Table 2). Following its comparison setup, we also adapted Crowd3D (CVPR'23) and GroupRec (ICCV'23), originally single-image crowd reconstruction methods, to video sequences. Additionally, we conducted both quantitative and qualitative comparisons with VideoMimic (CoRL'25), the state-of-the-art small-scene joint optimization method with publicly available code prior to our submission. As shown in Table 4 and Figure 5, small-scene methods cannot be directly applied to large-scale crowd reconstruction under the same settings. The field is still rapidly evolving, and we believe our work not only provides important technical contributions, but also serves as a foundational framework for 3D pseudo ground-truth annotation by combining with improved geometry reconstruction methods or real-world scans, with the potential to address the critical data scarcity challenge in 3D crowd datasets.
>
> **Q2. A more detailed explanation of slight metric changes in ablation experiments**
>
> Thank you for this observation. We provide the following explanations:
>
> Table 5 (Multi-stage ablation): The minimal change in PPDS from Stage-1 (88.76) to Full (89.04) is because Stage-1's HSIP already establishes strong global consistency in crowd distribution. The subsequent stages (Stage-2 and Stage-3) primarily improve local pose accuracy, as reflected by the significant MPJPE reduction (76.23 → 61.83), rather than global distribution metrics.
>
> Table 7 (CSCR ablation): This stems from different metric sensitivities. PPDS measures overall distribution consistency of large-scale crowds, making it relatively insensitive to local corrections. MPJPE directly evaluates individual joint accuracy. CSCR operates at the local neighborhood level, strengthening relative displacement consistency between adjacent individuals. In crowd scenarios, inaccurate root positions propagate to pose estimation, causing amplified MPJPE errors. CSCR optimizes local spatial relationships, significantly improving MPJPE while having minimal impact on global metrics like PPDS.

---

> > ### Author Rebuttal · Reviewer_4KCy · 2026-04-02
> >
> > I think W1 and W2 need to be supplemented with quantitative and qualitative results, rather than statements.
> >
> > Update: The authors have provided further results. I have no further questions, and I keep my score.

---

> > > ### Author Response · Authors · 2026-04-06
> > >
> > > **W1. The qualitative results of ablations are required**
> > >
> > > Thank you for your continued suggestion. We sincerely appreciate your feedback, which has been very helpful in improving our paper. We provide qualitative visualizations for ablation studies at the [anonymous link](https://anonymous.4open.science/r/icml-72B4/README.md). Specifically:
> > >
> > > (1) As illustrated in Fig. 1 of the anonymous link, removing the HSIP module leads to a clear mismatch between generated poses and the scene context, due to the absence of structural constraints imposed by HSIP. This demonstrates the necessity of HSIP for handling large-scale scene-level tasks.
> > >
> > > (2) Since the CSCR loss is built upon the group-structure prior introduced by HSIP, it serves as a complementary refinement mechanism. It stabilizes group-wise spatial arrangements while further refining positions and affecting subtle pose variations, as also reflected in Table 7 of the original paper. Although such fine-grained changes are difficult to clearly observe in crowded scenes, incorporating CSCR constraints better ensures the consistency between local groups and the scene. This effect is clearly illustrated in Fig. 2 of the anonymous link.
> > >
> > > **W2. The computational cost comparisons with other baselines are also required**
> > >
> > > Thank you for your continued feedback on this point. We provide detailed computational cost comparisons on VirtualCrowd Scene-1 in Table R2.
> > >
> > > Table R2. Computational cost comparison with baseline methods.
> > >
> > > | Method | PPDS↑ | PA-PPDS↑ | PCOD↑ | MPJPE↓ | ACCEL↓ | Runtime(min)↓ |
> > > |----------|-------|--------|--------|--------|--------|--------|
> > > | DyCrowd (other baseline) | 89.44 | 91.18 | 92.49 | 56.69 | **13.46** | 18+27+70+104=219 |
> > > | Crowd4D (Ours-Fast) | 90.98 | 91.51 | 94.05 | 55.25 | 14.21 | 18+16+35=69 |
> > > | Crowd4D (Ours-Full) | **91.08** | **91.59** | **94.13** | **53.61** | 13.56 | 27+26+139=192 |
> > >
> > > The runtime values are presented as the sum of time costs for different processing stages. The overall efficiency depends on the number of optimization iterations. Ours-Fast uses 100/100/100 iterations for the three stages respectively, while Ours-Full uses 150/150/300 iterations. As shown in the table, Crowd4D achieves lower runtime than DyCrowd while ensuring better accuracy. For applications prioritizing efficiency, Ours-Fast provides faster processing with acceptable accuracy trade-offs. All experiments were conducted on a single NVIDIA 4090D GPU. With multi-GPU parallelization, the actual processing time would be further reduced.

---

### Official Review · Reviewer_obqY · 2026-03-11

**Soundness:** 3
**Presentation:** 3
**Significance:** 3
**Originality:** 3
**Overall Recommendation:** 4
**Confidence:** 4

**Summary:**

This paper presents Crowd4D, a method for reconstructing 4D crowd motion in large-scale, complex terrain scenarios from monocular RGB videos. On the VirtualCrowd synthetic dataset, the proposed method surpasses the existing baseline DyCrowd.

**Compliance With Llm Reviewing Policy:**

Affirmed.

**Final Justification:**

Thank you for your response and for addressing my concerns.  After consideration, I will keep my original score unchanged.

**Key Questions For Authors:**

1. Clarify dataset split.
2. Quantify on PANDA.
3. Analyze failure modes

**Limitations:**

yes

**Strengths And Weaknesses:**

Strength: The paper addresses a highly relevant and challenging problem in computer vision: reconstructing dense, 4D crowds in large-scale, real-world environments with complex terrain. While the high-level idea of human-scene interaction constraints has been explored in works like HSC4D, this paper's originality lies in successfully adapting this concept to the much more difficult setting of monocular video. The proposed pipeline is insightful. The paper is generally well-written and organized.
Weakness:
1. The paper fails to specify how the VirtualCrowd dataset was split for training/validation/testing.
2. The validation on the real-world PANDA dataset is purely qualitative. While the visual results appear compelling, they are not backed by any quantitative metrics. The PANDA dataset provides ample 2D annotations, including 15 million bounding boxes and over 12,000 trajectories. Even without 3D pose ground truth, metrics like bounding box reprojection error could be calculated to provide a quantitative measure of the method's performance in the real world. This would significantly strengthen the claim that the method works beyond synthetic data.
3. The paper's core hypothesis—that two noisy signals can be combined to yield a robust constraint—is intuitively appealing but carries inherent risk. The paper does not deeply analyze scenarios where this assumption might break down. For instance, if the scene point cloud has a systematic bias (e.g., a consistently over-estimated slope) or if the initial SMPL fits are catastrophically wrong due to severe and persistent occlusion, the HSIP constraint could pull the solution towards a plausible but still incorrect local minimum. The lack of a detailed failure analysis or a discussion of potential limitations makes it difficult to assess the method's robustness in unforeseen, "in-the-wild" conditions.

---

> ### Author Rebuttal · Authors · 2026-03-31
>
> We thank Reviewer obqY for the positive assessment and for recognizing the originality and insightfulness of our pipeline. Below we address the raised concerns.
>
> **W1 & Q1. The paper fails to specify how the VirtualCrowd dataset was split for training/validation/testing**
>
> Thank you for raising this important question. We clarify that Crowd4D is an optimization-based method that does not involve learning or training procedures. Therefore, the VirtualCrowd dataset is used solely as a test set for evaluation purposes, not for training. This is consistent with the DyCrowd paper, which also uses VirtualCrowd for testing only, as large-scale crowd datasets with 3D annotations are currently unavailable. Notably, our optimization-based approach can generate pseudo-labels from real-world images, which can serve as supervision for developing learning-based methods.We will clarify this explicitly in the revised paper.
>
> We acknowledge that this is a fundamental limitation in the field. We consider the development of new large-scale crowd datasets with 3D annotations an important future direction for the community. We will explicitly clarify this point in the revised version to avoid any confusion.
>
> **W2 & Q2. The validation on the real-world PANDA dataset is purely qualitative**
>
> Thank you for this constructive suggestion. We clarify that while PANDA provides bounding box and trajectory annotations, it does not contain any pose-related data (neither 2D nor 3D pose annotations). Multi-object tracking is not the primary focus of our method, as we initialize from existing tracking results. Our optimization primarily targets joint-level 2D reprojection poses rather than overall bounding box positions, resulting in minimal changes to bbox-level metrics. Consequently, bounding box reprojection error is not meaningful for evaluating 3D reconstruction quality. Following the DyCrowd setting, we provide qualitative results on PANDA to validate real-world applicability. Notably, our method achieves improved reconstruction quality. Our optimization approach can be combined with scene geometry to extend 3D annotations for existing human-related datasets, directly addressing the critical data scarcity challenge. This will be a primary focus of our future work.
>
> **W3 & Q3. The paper's core hypothesis carries inherent risk and lacks detailed failure analysis**
>
> Thank you for this insightful observation. As shown in Table 6, without any geometric constraints, existing methods struggle to achieve unified crowd reconstruction in a shared scene due to individual scale estimation issues, let alone larger and more complex crowd scenarios. Our optimization strategy has preliminarily achieved this challenging task while also addressing the single-plane dependency of DyCrowd. We acknowledge that current scene reconstruction has limitations (e.g., multi-layer structures, incomplete SIS geometry, challenging visual conditions), which can affect reconstruction quality. However, the π³-based approach currently provides relatively stable results. With improved geometric reconstruction (better scene reconstruction methods or scanned scenes), our method can enable pseudo-label annotation for real-world video data, advancing the field.
>
> For individual reconstruction, our method not only relies on scene constraints but also achieves system-level robustness through crowd-level to individual-level motion prior constraints. This better addresses issues from initialization methods or occlusion. Table 3 demonstrates our performance on Occluded Instances. We will include detailed failure analysis in the revised version.

---

> > ### Author Rebuttal · Reviewer_obqY · 2026-04-02
> >
> > Thank you for your response and for addressing my concerns. I appreciate the clarifications provided. After consideration, I will keep my original score unchanged.

---

### Official Review · Reviewer_1Uz2 · 2026-03-14

**Soundness:** 3
**Presentation:** 3
**Significance:** 3
**Originality:** 3
**Overall Recommendation:** 4
**Confidence:** 4

**Summary:**

This paper presents Crowd4D, a scene-aware framework for monocular 4D crowd reconstruction in large-scale scenes with complex terrain. The key challenge is accurate human–scene alignment (scale and position) when both crowd and scene are recovered from monocular video. The authors introduce the Human–Scene Interaction Proxy (HSIP), built from Scene Interaction Point Clouds and a Scene Interaction Surface (SIPC&SIS), which encodes scene geometry to constrain each person’s feasible 3D region and stabilize metric scale. They also propose Crowd Structural Coherence Regularization (CSCR) to enforce soft temporal consistency of pairwise relative displacements and directions within local neighborhoods, improving stability under occlusions. A multi-stage optimization (scale/root, pixel-consistent pose, group-guided motion) jointly refines crowd and scene. Experiments on VirtualCrowd and PANDA show gains over DyCrowd and VideoMimic in PPDS, PCOD, MPJPE, and robustness on non-flat terrain and occluded cases.

**Compliance With Llm Reviewing Policy:**

Affirmed.

**Final Justification:**

Thank you for your reply. My concerns have been addressed.

**Key Questions For Authors:**

- SIS construction: How is the Scene Interaction Surface (SIS) obtained from the scene point clouds, e.g., Poisson or ball-pivoting mesh, implicit surface such as NeuS, or other? What happens when geometry is very noisy or incomplete, e.g., sparse keyframes? A short description or pointer to implementation would aid reproducibility. Understanding failure modes would affect the soundness assessment.

- Failure cases: When does HSIP-based anchoring fail, e.g., persons on stairs, overhangs, reflective ground, or when SIS has large holes? If you can provide 1–2 qualitative failure examples or a short discussion of scenarios to avoid, that would clarify the scope of the method and could improve the significance rating.

- Comparison to depth-prior methods: How does Crowd4D compare when using the same scene/depth backbone as VideoMimic or JOSH on the same large-scene data? If gains persist, that would support that HSIP/CSCR are the main drivers. If not, it would clarify the role of the full pipeline, e.g., $\pi^3$ vs. other scene modules.

**Limitations:**

The authors include an Impact Statement noting positive uses (urban design, safety, transportation) and privacy (no facial detail reconstruction, compliance with regulations). Suggest briefly discussing: (1) dependence on quality of monocular scene reconstruction and tracking, and (2) scenarios where SIS/HSIP may be unreliable, e.g., sparse geometry, reflective surfaces.

**Strengths And Weaknesses:**

### Strengths

- The evaluation is convincing: clear gains over DyCrowd on VirtualCrowd (PPDS, MPJPE, etc.) under both tracking settings, and ablations back up the multi-stage setup, the choice of HSIP over simpler anchors, and the role of CSCR.
- The HSIP construction (reference position, ray-space correction, vertical mapping to SIS) is explained in enough detail to follow the idea, and the move beyond a single ground plane to real scene geometry is a real step forward for monocular crowd work.
- The paper is well structured, and related work and the gap, e.g., single-plane assumptions, sensitivity of HVIP-style anchoring, are clear, and Figure 2 makes the pipeline easy to grasp.
- The overall idea, coupling crowd and scene via a compact interaction proxy and adding group-level temporal regularization, is novel and well motivated, and the results on non-flat terrain and occluded cases show practical benefit.

### Weaknesses

- Runtime is heavy, about 4 hours for 100 people $\times$ 200 frames, and it would help to know where the bottleneck lies and whether a lighter mode is feasible.
- Generally, the proposed Crowd4D is a complex system. More analysis on each component and failure cases will be helpful, e.g., ensitivity to tracking or scene reconstruction quality, or different reconstruction / initialization methods.
- The discussion of SynCHMR in Related Work is problematic. It does not assume static scenes.

---

> ### Author Rebuttal · Authors · 2026-03-31
>
> We thank Reviewer 1Uz2 for the constructive feedback and for recognizing our contributions on scene-aware 4D crowd reconstruction. Below we address the raised Weaknesses and Key Questions.
>
> **W1. Runtime is heavy**
>
> The bottleneck is the three-stage optimization (150/150/300 iterations), where Stage-3 accounts for ~2/3 of runtime due to its larger parameter space. A lighter mode is feasible: Table 5 shows Stage-1 already achieves strong consistency (PPDS=88.76), while later stages mainly refine pose accuracy. Running only the first two stages reduces runtime to under 1 hour with moderate accuracy trade-off. Multi-GPU parallelization could further reduce runtime. We will include a per-stage runtime breakdown in the revision.
>
> **W2. More analysis on each component and failure cases**
>
> Regarding sensitivity to scene reconstruction: while performance varies with reconstruction quality, this dependency is unavoidable for scene-aware methods. Our method achieves robustness through joint constraints from multiple energy terms. Regarding tracking: as shown in Figure 4, even with unified tracking-by-detection, our joint optimization enables better recovery from detection errors than DyCrowd. We will include additional component-wise analysis. Below is a comparison under different scene reconstruction backbones (DA3 and π³) on synthetic scene1:
>
> | Method | PPDS↑ | PA-PPDS↑ | PCOD↑ | MPJPE↓ | ACCEL↓ |
> |----------|-------|--------|--------|--------|--------|
> | DyCrowd | 89.44 | 91.18 | 92.49 | 56.69 | **13.46** |
> | Crowd4d w/ DA3 | 90.12 | 90.33 | 94.13 | 55.43 | 13.49 |
> | Crowd4d w/ π³ | **91.08** | **91.59** | **94.13** | **53.61** | 13.56 |
>
>
> The results show stable performance across scene reconstruction backbones, with π³ providing slightly better geometry.
>
> **W3. The discussion of SynCHMR in Related Work is problematic**
>
> Thank you for pointing this out. We have corrected Sec. 2.1: SynCHMR is now described as a SLAM/SfM-enhanced human-scene reconstruction method, and we clarify that our work targets large-scale crowd reconstruction with dense pedestrians, complex terrain, and severe depth ambiguity.
>
> **Q1. How is the Scene Interaction Surface (SIS) obtained from the scene point clouds?**
>
> We sample high-confidence points and select the lowest points within planar grids, then apply Poisson reconstruction to obtain SIS as a terrain-oriented surface abstraction. We will clarify this in Sec. 3.2.1 of the revision. Our goal is stable ground support for human anchoring rather than visual completeness. When local geometry is imperfect, system-level robustness is achieved through complementary energy constraints to mitigate errors caused by scene geometry. Failure modes are discussed in Q2. We will add implementation details in the revision and release the code.
>
> **Q2. When does HSIP-based anchoring fail?**
>
> Most crowd scenario research focuses on open environments where temporal sequences provide sufficient geometric information. We acknowledge that challenging cases—multi-layer structures, scene holes, and complex visual conditions—may cause geometric errors, which remain open challenges for the field and the broader community. However, since our method does not rely solely on scene geometry constraints, it can handle most noise-induced reconstruction issues: HSIP defines feasible regions as a soft constraint; multi-stage optimization allows corrections via image evidence and motion priors; CSCR stabilizes individual trajectories through group-level consistency. Limitations in complex environments remain, and failure examples will be included in the supplementary.
>
> **Q3. How does Crowd4D compare when using the same scene/depth backbone as VideoMimic or JOSH?**
>
> In fact, Table 4's comparison with VideoMimic already adopts the same scene reconstruction framework. Gains persist (PPDS: 91.43 vs 80.03, MPJPE: 59.35 vs 62.49, ACCEL: 12.93 vs 23.80), indicating improvements stem from our system-level optimization and targeted design for large-scale scenes via HSIP anchoring and CSCR, rather than backbone choice. JOSH shares similar depth priors and energy formulation with VideoMimic, so comparable results are expected. We will provide more comparisons with both methods in the revision.

---

### Decision · Program_Chairs · 2026-04-30

**Decision:**

Accept (regular)

**Comment:**

This paper proposes Crowd4D, a scene-aware framework for monocular 4D crowd reconstruction in large-scale scenes. The main idea is to employ a Human–Scene Interaction Proxy (HSIP) to couple scene geometry with crowd placement on non-planar terrain, as well as use Crowd Structural Coherence Regularization (CSCR) to enforce temporal stability under occlusions. All reviewers acknowledged that the problem is well-motivated and practically meaningful, and the proposed pipeline seems reasonable. The main concerns, however, include higher inference cost, missing qualitative ablations, and limited evaluation on the real-world PANDA dataset. In the rebuttal, authors provided a Fast mode (69min) that outperforms DyCrowd (219min) in both efficiency and accuracy, as well as included qualitative ablation visualizations that demonstrate the distinct roles of HSIP and CSCR. Reviewers obqY and 4KCy confirmed all concerns resolved but did not raise the final rating. Reviewer 1Uz2 did not provide a formal acknowledgement but provided a short final justification to acknowledge that their concerns have been addressed, and Reviewer C3iu did not actively participate in the discussion and did not change the original rating. Overall, it seems that all reviewers remain lukewarm of the work and AC leans slightly towards accept.